# Creation of a Corneal Flap for Laser In Situ Keratomileusis Using a Three-Dimensional Femtosecond Laser Cut: Clinical and Optical Coherence Tomography Features

Antonio Leccisotti [1,2,3,*], Stefania V. Fields [1,3], Giuseppe De Bartolo [1,3], Christian Crudale [1,3] and Matteo Posarelli [1,3,4]

1 Siena Eye Laser, 53036 Poggibonsi, Italy; stefania.fields@virgilio.it (S.V.F.); giusedebartolo@gmail.com (G.D.B.); christiancrudale@gmail.com (C.C.); mposarelli@gmail.com (M.P.)
2 School of Biomedical Sciences, Ulster University, Coleraine BT52 1SA, UK
3 Centre for Research in Refractive Surgery, 53035 Siena, Italy
4 Department of Ophthalmology, Liverpool University, Liverpool L3 5TR, UK
* Correspondence: leccisotti@libero.it; Tel.: +39-335-8118-324

**Abstract:** Laser in situ keratomileusis (LASIK) is the most frequently used technique for the surgical correction of refractive errors on the cornea. It entails the creation of a superficial hinged corneal flap using a femtosecond laser, ablation of the underlying stromal bed using an excimer laser, and repositioning of the flap. A corneal flap with an angled side cut reduces the risk of flap dislocation and infiltration of epithelial cells and confers unique biomechanical properties to the cornea. A new laser software creating three-dimensional (3D) flaps using a custom angle side cut was retrospectively evaluated, comparing optical coherence tomography 3D (with intended 90° side cut) and 2D flaps (with tapered side cuts) as well as respective intra- and early postoperative complications. Four hundred consecutive eyes were included, two hundred for each group. In the 3D group, the mean edge angle was 92°, and the procedure was on average 5.2 s slower ($p = 0$). Non-visually significant flap folds were found in thirteen eyes of the 2D group and in seven eyes of the 3D group ($p = 0.17$). In conclusion, the creation of a LASIK flap using a 3D femtosecond laser cut, although slightly slower, was safe and effective. The side cut angle was predictable and accurate.

**Keywords:** femtosecond laser; optical coherence tomography; laser in situ keratomileusis

## 1. Introduction

The correction of refractive errors (myopia, astigmatism, and hyperopia) based upon reshaping of the corneal profile can currently be achieved via various techniques [1]; the most used is laser in situ keratomileusis (LASIK), in which a hinged corneal flap is created using a femtosecond laser, the flap is folded, and the underlying stromal bed is photoablated using a excimer laser to change its curvature, thus modifying its refractive power. The flap is then replaced and allowed to adhere via oncotic pressure [2].

The edges (side cuts) of the corneal flap can be tapered or angled, with the latter shape having several advantages: ease of repositioning during surgery, resistance to tangential forces due to eyelid movement (hence, resistance to early dislocation) [3], stronger adhesion via scarring, and barrier effect against the infiltration of epithelial cells in the interface (epithelial ingrowth) [4]. Precise side cut angles induce different biomechanical properties in the cornea [5,6]. However, in some laser platforms the effective angle of the side cut may considerably differ from the programmed angle, even by 45°, and with a high standard deviation [7], thus rendering the induced benefits unpredictable.

The femtosecond laser Ziemer Z8 (Ziemer Group, Port, Switzerland) can create a LASIK flap on the applanated cornea with a contact flat glass fixated via scleral suction, through which the laser is delivered at a defined depth. In the classic two-dimensional

(2D) method, the flap size is determined by the applanated area, and its edges are defined by the border of the applanated cornea, resulting in a tapered shape (Figures 1–3). In the new three-dimensional (3D) application, released on 1 February 2024, a side cut is actively created at the desired angle, so that the flap size and edges are planned on a dedicated software (Figures 4 and 5). There has not, however, been a study to verify the effective side cut angle achieved using this new application nor the possible additional time under suction required to perform a 3D flap.

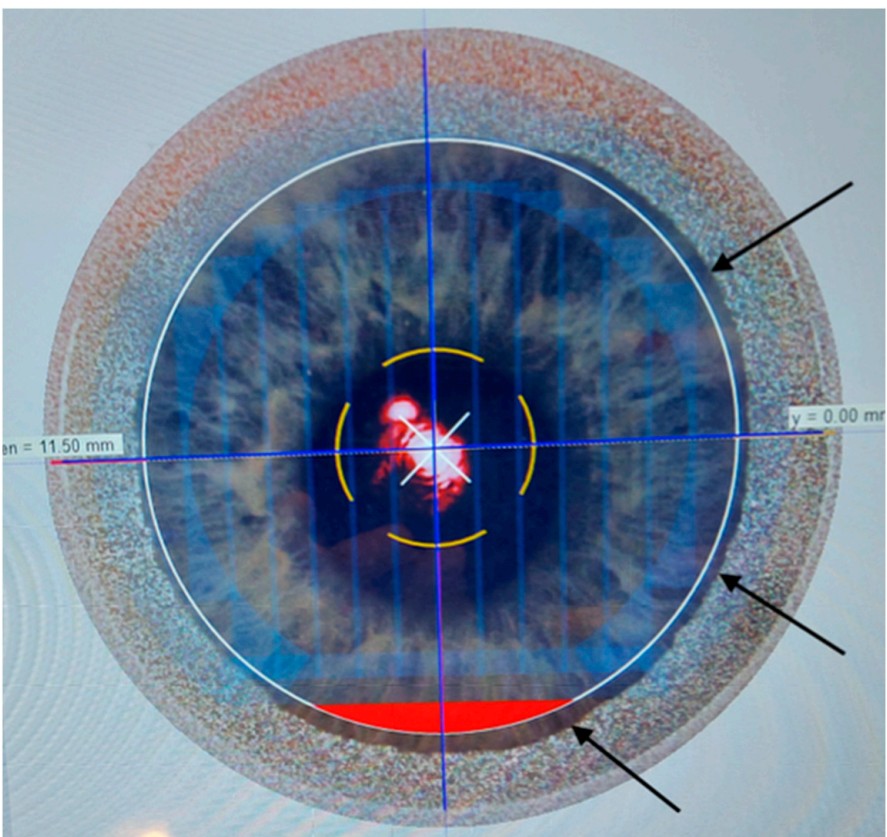

**Figure 1.** Computer screen of the femtosecond laser Ziemer Z8 (Ziemer Group, Port, Switzerland) in the early phase of the creation of a LASIK flap using the 2-dimensional program (surgeon view, with inverted image). The central red reflex is the aiming beam applied to the patient. The red inferior crescent is the hinge side. The white circle is the planned flap size. The black arrows indicate the peripheral edge of the cornea applanated by the flat glass.

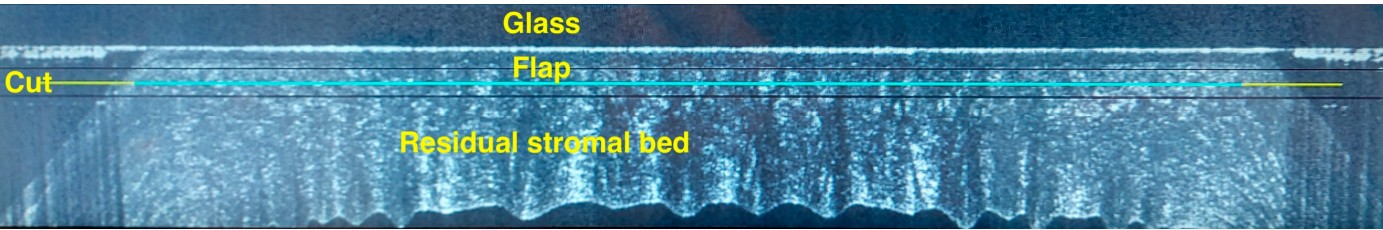

**Figure 2.** Same case as in Figure 1: intraoperative optical coherence tomography (OCT) during scleral suction and before a 2-dimensional laser cut, showing the laser glass applanating the cornea and the programmed cut (blue line) to divide the superficial flap from the residual stromal bed. The corneal posterior surface shows compression folds. The horizontal cut on the applanated corneal surface causes tapered flap edges.

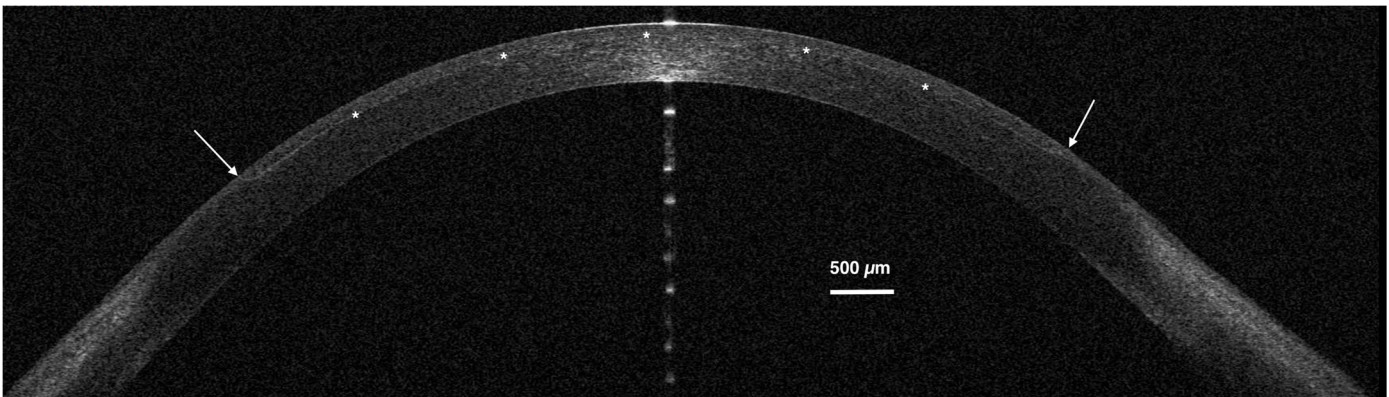

**Figure 3.** Same case as in Figure 1: right eye, postoperative optical coherence tomography (OCT) at day 1, horizontal corneal section. After a 2-dimensional cut, the flap edges (arrows) are slightly curved. The asterisks indicate the interface created by the laser between the flap (external) and the residual stromal bed (internal).

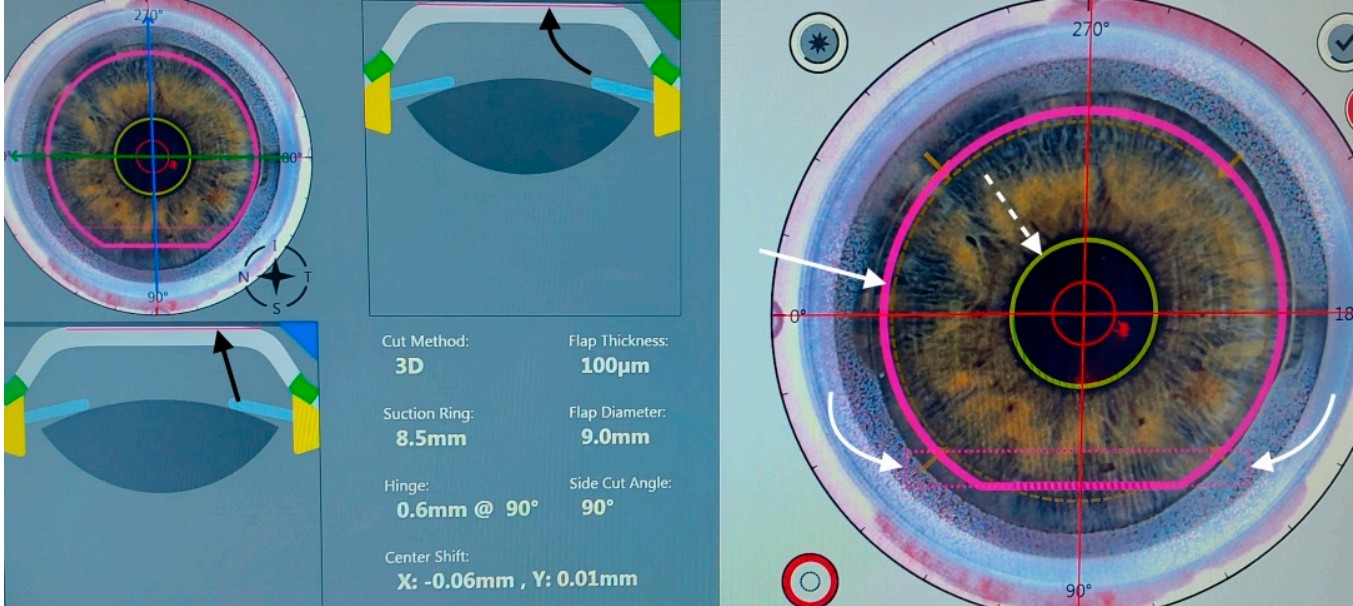

**Figure 4.** Computer screen of the femtosecond laser Ziemer Z8 (Ziemer Group, Port, Switzerland) in the early phase of the creation of a LASIK flap using the 3-dimensional program (surgeon view, with inverted image). The purple line (straight white arrow) indicates the planned flap size, with a superior hinge. The screen displays all the flap settings. The yellow circle (dotted white arrow) corresponds to the pupil, as recognized by the system. The straight black arrow indicates the planar cut on the vertical meridian; the curved black arrow points to the planar cut on the horizontal meridian. The 2 white curved arrows indicate the exits of the 2 venting channels, delimited by the pink dotted box.

Anterior segment optical coherence tomography (OCT) is the ideal imaging technique to depict the corneal details after laser surgery [7,8]; we, therefore, compared 2D- and 3D-generated LASIK flaps in a large series using the new software, to assess the (1) mean time of suction, (2) mean effective angle of the side cut on OCT, and (3) intraoperative and early postoperative complications.

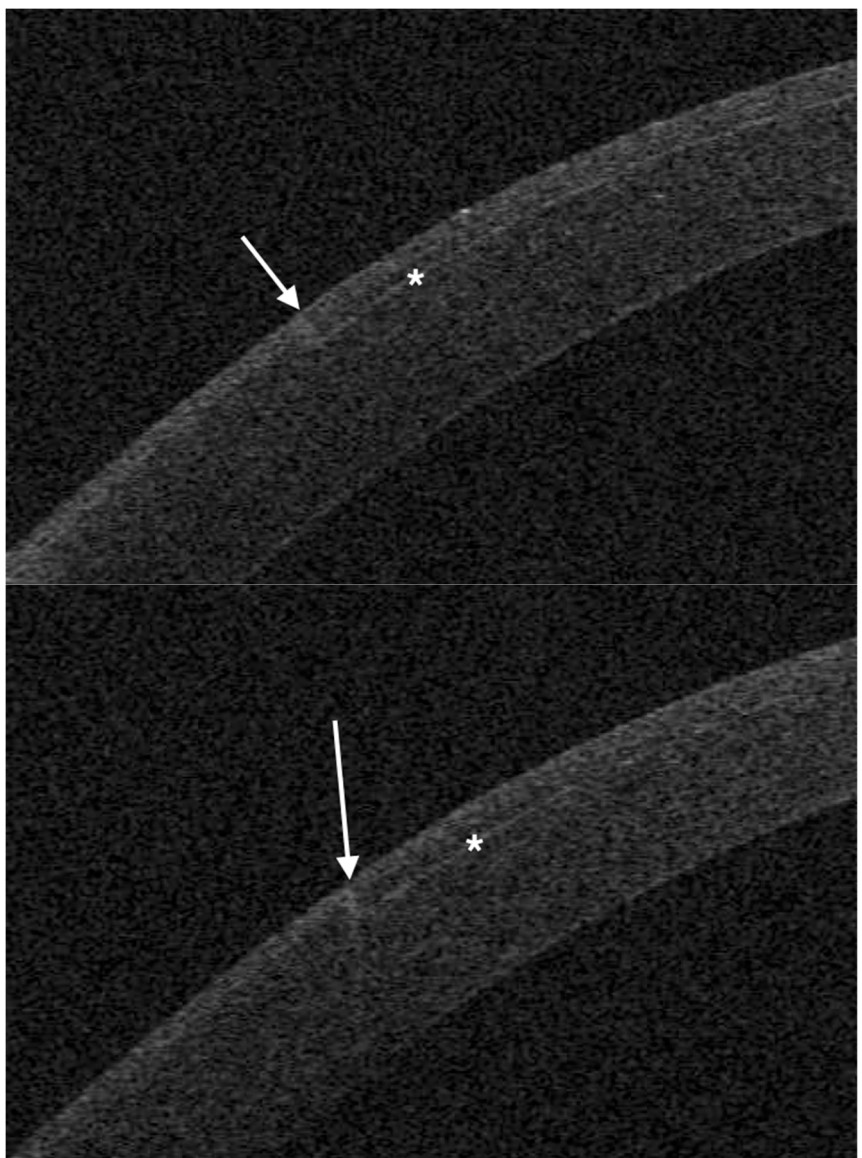

**Figure 5.** Postoperative optical coherence tomography (OCT) at day 1; detail of horizontal corneal section after a 3-dimensional cut using a programmed 90° side cut (**top image**) and a 110° side cut (**bottom image**). The arrow indicates the side cut; the asterisk represents the interface created by the laser between the flap (external) and the residual stromal bed (internal).

## 2. Materials and Methods

A retrospective, comparative case series study was designed, including consecutive myopic patients who had undergone femtosecond LASIK in our institute between February and March 2024. In patients receiving bilateral treatment, a single eye was randomized to be included. The Institutional Review Board (Commissione di Revisione dell'Istituto Siena Eye Laser) provided approval on 1 March 2024 (code 07/2924). The research followed the tenets of the Declaration of Helsinki; patients provided a signed informed consent.

Inclusion criteria for surgery were as follows:

(a) Myopia or compound myopic astigmatism with spherical equivalent (SE) −1 to −12 diopters (D), with at least 18 months of refractive stability, and a refractive astigmatism ≤ 3 D.
(b) Age: between 25 and 50 years;
(c) General health status: absence of collagen vascular disease; no pregnancy;
(d) Ocular disease: no previous surgery; absence of scars or epithelial irregularities; absence of macular or lens abnormality; no topical treatment for ocular hyperten-

sion; absence of dry eye symptoms; non-invasive tear film break-up time $\geq 10$ s (MS-39, Costruzione Strumenti Oftalmici, Florence, Italy); lacrimal fluid osmolarity $\leq 300$ mOsm/l (I-PEN, Imedpharma, St. Laurent, QC, Canada);

(e) Corneal features on OCT and Placido topography (MS-39): central pachymetry $\geq 480$ μm; regular posterior elevation; anterior and posterior tangential topography; no signs of ectasia;

(f) Corrected distance visual acuity (CDVA) $\geq 20/40$ Snellen;

(g) Minimum follow-up: 1 month from treatment.

The preoperative assessment consisted of uncorrected distance visual acuity (UDVA), CDVA, manifest and cycloplegic refraction (by tropicamide eye drops), undilated and dilated slit-lamp evaluation, Placido corneal topography, OCT tomography with epithelial and stromal thickness evaluation, computer-assisted scotopic pupillography, tonometry, and tear function evaluation.

Soft contact lens use was interrupted 1 month before examination and surgery; rigid contact lens use was interrupted 3 months before examination and surgery. All patients were informed about the surgical procedure and provided written consent.

Treatment allocation was determined via the exclusive use of 3D software since 8 February 2024: therefore, 200 eyes of the last 200 consecutive patients treated until 7 February 2024 formed the 2D group. The 3D group was then formed by 200 eyes of the first 200 consecutive patients treated since 8 February 2024.

Our technique for femtosecond LASIK has been described in detail [9]. In brief, full manifest spherical and cylindrical corrections were set. The planned thickness of the flap was 100 μm in both groups. In all cases, it was programmed to leave a stromal bed thickness $\geq 300$ μm. The suction ring diameter was chosen according to horizontal corneal diameter measured using OCT. Femtosecond laser power and velocity were adjusted to obtain a uniform pattern of tiny, non-confluent plasma bubbles. After topical anaesthesia with oxybuprocaine, a drop of unpreserved 0.2% sodium hyaluronate was dripped on the cornea, and the applanation glass was applied to the cornea; centration was initiated as soon as scleral suction was activated.

The new 3D software is characterized by the possibility of designing all the parameters of the LASIK flap, including thickness, shape, diameter, angle of side cuts, location, and length of the hinge. In addition, it can automatically centre the flap on the pupil and align the flap axis on limbal marks provided by a coloured surgical marker. Intraoperative OCT can be activated to verify that the intended flap area is in the correct position. To avoid a temporary intrastromal accumulation of plasma (opaque bubble layer, OBL), 2 ventilation channels were created at the base of the hinge. All these features are unique to this software and have no correspondence to the other femtosecond platforms. The Z8 femtosecond laser has a wavelength of 1035 nm, a pulse energy of 0.15 mJ, a repetition rate of 20 MHz, a flat applanating surface, and a raster cut pattern.

Flap centration on the cornea can be achieved by addressing patient voluntary eye movements towards a reference aiming light and then adjusting the centration using arrows on the laser touchscreen. The new 3D software includes a self-centration option, where the pupil is automatically recognized and the flap is centred on it. In all 3D-treated eyes, the self-centration option was used. After centration, the femtosecond treatment was initiated and the total time under activated suction was measured.

After the completion of the femtosecond phase, the flap was separated and folded in a "taco" fashion using a flap spatula (MMSU1171, Malosa Surgical, Halifax, UK). After the refractive treatment was conducted using a Teneo 317 excimer laser (Bausch+Lomb, Bridgewater, NJ, USA) in Planoscan mode, the flap was repositioned, the interface was washed with balanced salt solution for 2 s through a single use 25-gauge cannula, and the flap was finally smoothed down using a wet microsponge. A drop of unpreserved netilmicin 0.3% + dexamethasone 0.1% was dripped on the cornea. Netilmicin and dexamethasone were continued 4 times daily for a week.

Only eyes where a 9 mm flap was planned (using the suction ring diameter with the 2D setting, with the laser setting on 3D) were included. In the 3D group, a 90° angle was chosen for the side cut.

OCT was performed the day after surgery with a horizontal high-resolution section; the side cut angle was measured on the tangent line of the corneal curvature at the temporal aspect of the cornea using the MS-39 software tool.

Statistical analysis was performed using the SPS software, available online at www.statisticsfordataanalysis.com (accessed 7 March 2024). The mean ± standard deviation was used to describe quantitative variables, and a *p* value less than 0.05 was considered statistically significant.

## 3. Results

A total of 400 eyes were finally included, 200 for each group. Table 1 reports the preoperative features of the two groups and the outcome measures. Mean age, male/female ratio, right/left eye ratio did not differ significantly between groups. All patients were Caucasian.

**Table 1.** Preoperative data in 2 groups of 200 eyes undergoing laser in situ keratomileusis (LASIK) with flap creation via 2-dimensional and three-dimensional femtosecond laser cuts. Values (±standard deviation).

| Parameter | 2D | 3D | *p* Value (95% CI) |
|---|---|---|---|
| Age (years) | 36.4 (±4.5) | 37.2 (±6.1) | 0.27 [§] (−1.85 to 0.25) |
| Male/female (% males) | 74/126 (37) | 68/132 (34) | |
| Right/left eye (% right) | 94/106 (47) | 85/115 (43) | |
| Suction time (s) | 31.1 (±3.3) | 36.3 (±4.2) | 0 [§] * (−5.9 to −4.6) |
| Angle of sidecut (degrees) | 21 (±5) | 92 (±3) | |
| Opaque bubble layer | 2 | 9 | 0.03 [^] * |
| Folds | 13 | 7 | 0.25 [^] |
| Diffuse lamellar keratitis | 1 | 0 | |

[§]: p value of 2-tailed *t*-test for unpaired data; [^]: p value of Chi$^2$ test of independence; 95% CI: 95% confidence interval for the difference between means; *: statistically significant, with $p < 0.05$.

The mean suction time was 31.1 s (standard deviation, SD 3.3) in the 2D group and 36.3 s (SD 4.2) in the 3D group; the difference between the means (5.2 s) is statistically significant ($p = 0$).

Irregular edges or difficult dissection were not encountered in either group.

An OBL was observed in two cases (two peripherals, none central) after 2D treatment and in nine cases (six peripherals, three in the flap area) after 3D treatment; the difference is statistically significant ($p = 0.03$ in Chi$^2$ test of independence; Chi$^2$ value 4.58). The OBL did not interfere with the completion of the treatment in the 3 cases in any case (Figures 6 and 7).

The mean angle of the temporal edge of the flap was 21° (SD 5°) in the 2D group and 92° (SD 3°) in the 3D group.

Visually non-significant flap folds at day 1 were found in thirteen eyes post 2D treatment and in seven eyes post 3D treatment ($p = 0.25$ in Chi$^2$ test of independence; Chi$^2$ value 1.89).

Interface inflammation (diffuse lamellar keratitis) was encountered only in one eye in the 2D group, at grade 1 (peripheral), resolving with topical steroids in 2 weeks without complications.

No flap dislocations occurred in either group.

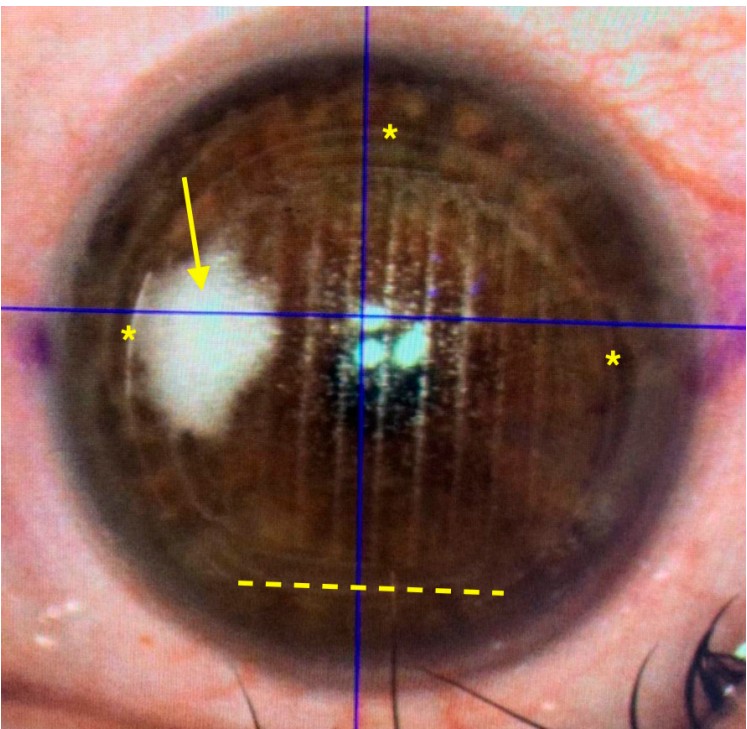

**Figure 6.** Intraoperative image (inverted surgeon view) soon after the creation of a LASIK flap using a 3-dimensional femtosecond laser cut, showing the formation of an opaque bubble layer (OBL) (arrow) in the flap interface. The asterisks indicate the flap edge; the yellow dotted line is the flap hinge. The parallel lines within the flap reflect the laser photodisruption pattern.

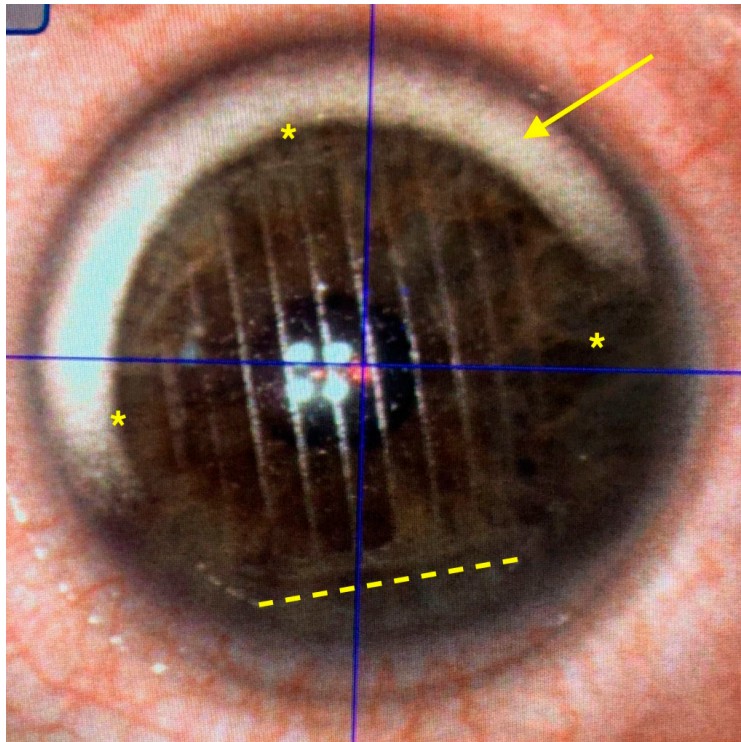

**Figure 7.** Intraoperative image (inverted surgeon view) soon after the creation of a LASIK flap using a 3-dimensional femtosecond laser cut, showing the formation of an opaque bubble layer (OBL) (arrow) outside the flap. The asterisks indicate the flap edge; the yellow dotted line is the flap hinge.

## 4. Discussion

In our series, the use of the new 3D software allowed a precise flap cut using femtosecond LASIK, with no complications. The only significant differences compared to the 2D program were a slightly increased occurrence of an OBL and an increased suction time (5.2 s longer). The flap edge was, as programmed, angled at 90°. A non-significant tendency to reduce flap folds was also observed in the 3D group.

The advantages of a 3D side cut have been described by previous studies and include resistance to dislocation and trauma, stronger marginal adhesion, easier surgical reposition of the flap, and less epithelial ingrowth [3,4,10]. The effects of different side cut angles on dryness, light sensitivity, and foreign body sensation are not significant [11]. A 90° side cut angle decreases the mechanical strain on the wound, compared to 30° [5]; biomechanical properties are, however, similar between different side cut angles [6]. The degree of marginal inflammation and scar formation is not influenced by the side cut angle [12] and neither is visual acuity [13].

The angle of the side cut achieved using femtosecond LASIK has only been evaluated in a study with another laser platform; the effective side cut angle diverged significantly from the programmed angle, with a mean difference of 25° [7]. In our series, the achieved angle was, instead, both accurate and precise.

The importance of the suction duration is determined by the rise in intraocular pressure (IOP), which may damage the eye. During suction with the Ziemer laser, the IOP reached 184 mmHg in an animal experimental model [14]; acute IOP increase can induce optic neuropathy and retinal detachment [15–17]. However, IOP rise is probably attenuated by the volume of blood in eye vasculature, displaced by the increased IOP [18]. Is unknown whether a 5-second difference poses a further risk of complication; in our series, no complications caused by suction were observed in either group.

An OBL is induced by the photodisruption caused by the femtosecond laser in the cornea, generating plasma composed of water vapour and carbon dioxide [19]. A dense OBL may interfere with the subsequent excimer laser phase [20]. Different patterns of treatment during femtosecond flap creation may influence the formation of an OBL, as the plasma may escape earlier and laterally from the applanated area, thus avoiding accumulation [21]. The occurrence of a central OBL covering the pupil may affect the centration tracking systems and, therefore, require a delay in treatment until the gas is reabsorbed. In the 2D program, the peripheral part is treated first, thus creating an escape route for the plasma during the whole procedure. In the 3D program, two vents are created near the hinge at the beginning of the treatment, but the flap edge is only completed at the end of the procedure, so plasma may still unduly accumulate. The cases of OBL that we observed were not central, not posing any challenge to the completion of the LASIK procedure.

Flap folds (striae) are a common complication of LASIK, that, in some, cases may become visually significant; flap malposition during surgery is one of the causes [22]. The cases in our series were mild and did not require treatment; although non-statistically significant, a slight reduction in their incidence was seen in the 3D group. A sharp flap edge may contribute to a better intrasurgical alignment.

Early flap dislocation after LASIK usually occurs spontaneously, without traumatic events; its incidence is 0.012% and higher with meniscus-shaped flaps [3]. Our sample is too small to compare its occurrence in the two groups, due to the rare occurrence of early flap dislocation. Late flap dislocation is usually traumatic and often caused by a tangential trauma [23]; an angled flap edge may theoretically provide more solid cicatricial adhesion and less susceptibility to tangential traumas.

Epithelial ingrowth is caused by corneal epithelial cells either introduced during surgery or infiltrating through the flap edge [24]. In an experimental model, angled flap edges reduce its incidence by blocking epithelial cell migration [4].

The limitations of the present study include the retrospective design and the relative rarity of possible complications. This study was, however, conceived to assess the safety of the surgical procedure and the effective shape of the flap using the new 3D software.

### 5. Conclusions

In conclusion, the creation of a LASIK flap using a 3D femtosecond laser cut was safe and effective, and the angle of the side cut was precisely achieved. Long-term studies are required to evaluate the impact of an angled flap edge on epithelial ingrowth and spontaneous or traumatic flap dislocations. The different profile of the side cut must be also evaluated in terms of induced optical aberrations, as in a previous study [12]. Other perspectives include the long-term assessment of flap adhesion and resistance to trauma and the effect of a different side cut in cases of retreatments. The biomechanical effects of different angles must also be compared on the grounds of biomechanical effects.

**Author Contributions:** Conceptualization, A.L., S.V.F. and G.D.B.; Methodology, A.L. and S.V.F.; Software, C.C.; Validation, G.D.B., M.P. and C.C.; Formal Analysis, A.L. and M.P.; Investigation, A.L.; Resources, M.P.; Data Curation, S.V.F. and C.C.; Writing—Original Draft Preparation, A.L. and M.P.; Writing—Review and Editing, A.L., S.V.F., G.D.B., C.C. and M.P.; Visualization, S.V.F.; Supervision, A.L. All authors have read and agreed to the published version of the manuscript.

**Funding:** This research received no external funding.

**Institutional Review Board Statement:** The study was conducted in accordance with the Declaration of Helsinki, and approved by the Institutional Review Board of Siena Eye Laser (protocol code 07/2024, 1 March 2024).

**Informed Consent Statement:** Informed consent was obtained from all subjects involved in the study. Written informed consent has been obtained from the patients to publish this paper.

**Data Availability Statement:** The raw data supporting the conclusions of this article will be made available by the authors on request.

**Conflicts of Interest:** The authors declare no conflicts of interest.

### Abbreviations

| | |
|---|---|
| CDVA | corrected distance visual acuity |
| LASIK | laser in situ keratomileusis |
| OBL | opaque bubble layer |
| OCT | optical coherence tomography |
| UDVA | uncorrected distance visual acuity |

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
