# Peer review of "Creation of a Corneal Flap for Laser In Situ Keratomileusis Using a Three-Dimensional Femtosecond Laser Cut: Clinical and Optical Coherence Tomography Features"

_optics, doi:10.3390/opt5020019_

Round 1
Reviewer 1 Report
Comments and Suggestions for Authors
In this manuscript, the authors created a LASIK flap by a 3D-femtosecond laser cut and evaluated its safety and efficacy. I have some comments that need to be addressed before the manuscript can be considered for publication in this journal:
1. The authors administered an increased suction time (5.2 seconds longer) in their protocol, and they also cited a reference that showed the IOP can reach 184 mmHg in an experimental porcine model during suction with the Ziemer laser. What is the mechanisms that an increased suction time can cause the elevation of IOP? How about the IOP values using human eye in this report? 184 mmHg of IOP is extremely high (around 10 times higher compared with normal IOP in human), it raises some safety concern, we all know that IOP of 40-50 mmHg in human can cause huge issues such as retinal vascular occlusion.
2. The figures are of high quality, however, some figures need scale bars.
3. The authors used 400 eyes of patients and controls, the ages and genders were mentioned in this report. How about the race in these patients or controls?
There are so many abbreviations in this manuscript, if possible, the authors can add an “Abbreviation or Acronym” section.
Thank you!
Author Response
REPLY TO REVIEWS
Dear Editor and Reviewers,
Thank you very much for your help and input. Our answers are in italics.
Best regards
The authors
REVIEWER 1
In this manuscript, the authors created a LASIK flap by a 3D-femtosecond laser cut and evaluated its safety and efficacy. I have some comments that need to be addressed before the manuscript can be considered for publication in this journal:
- The authors administered an increased suction time (5.2 seconds longer) in their protocol, and they also cited a reference that showed the IOP can reach 184 mmHg in an experimental porcine model during suction with the Ziemer laser. What is the mechanisms that an increased suction time can cause the elevation of IOP? How about the IOP values using human eye in this report? 184 mmHg of IOP is extremely high (around 10 times higher compared with normal IOP in human), it raises some safety concern, we all know that IOP of 40-50 mmHg in human can cause huge issues such as retinal vascular occlusion.
We would like to emphasize that this is a retrospective study on an already existing femtosecond laser, with CE mark and a history of 25 years of use and millions of treatments performed. We are not experimenting a new feature of it: it is just an evaluation of a detail in the creation of the LASIK flap side cut. The 3D design requires 5 seconds more, not an increase in the IOP compared to standard treatment. It is however estimated that, in vivo, IOP rise is less, due to the volume occupied by blood vessels (Strohmaier, 2013, DOI:10.1167/iovs.12-11155).
- The figures are of high quality, however, some figures need scale bars.
A scale bar has been added to the first OCT image (figure 3).
- The authors used 400 eyes of patients and controls, the ages and genders were mentioned in this report. How about the race in these patients or controls?
All patients are Caucasian. This was now added (page 8, line 142)
There are so many abbreviations in this manuscript, if possible, the authors can add an “Abbreviation or Acronym”
section.
An Abbreviations section has been added (page 11).
Reviewer 2 Report
Comments and Suggestions for Authors
The article “Creation of the corneal flap for laser in situ keratomileusis with a 3-dimension femtosecond laser cut: clinical and optical coherence tomography features” is an early study to validate and compare the new 3D refractive software of the Ziemer Z8 laser in a retrospective cohort.
The study’s aim/novelty is to collect preliminary evidence on the new program for use of flap design in order to facilitate higher powered, prospective studies in the future. This can generally be considered a legitimate reason to conduct a clinical study. Please make the research hypothesis and questions more clear in the present manuscript.
It is evident from the language used that the authors have committed to a surgical technique and technical system. Whether it is adequate to promote their technique (call it “goldstandard”) and feature the technical system is to some degree left in the responsibility of the authors.
The reporting of the study could be improved in terms of organisation and logic, please consider relevant reporting guidelines. The images appear a bit isolated in the manuscript; it would be better to group and arrange relevant images in a logic way as a panel. They could also be made a bit smaller.
It is recommended that the refractive software which seems important for the present study is better described. Specifically, the following questions should be answered: When was the 3D software released? What are its capabilities (including novelties compared to other suites on the market)? How does the updated version differentiate from previous versions (such as the 2D model, optima)? It would be good to illustrate the software with a telling screenshot, image (or even video) which demonstrates its functionalities. In addition, is the capability of the software really unique or can manual techniques accomplish a similar result? It is important to link the functionality of the software to the results achieved in the present clinical study (how does the software produce these results?). This is very central if the aim is to conduct future studies on the topic. The output of the present research project should be the generation of new hypothesis which subsequently can be addressed in follow-up studies in an improved research setting. Please specify these newly generated hypothesis to generate a basis for future research.
On a technical/methodological level, would it be interesting to follow the cases with aberrometry or are the results unlikely to be different from other surgical approaches?
Refractive surgeons often have relationships to the industry due to the technical nature of the discipline which requires the question of a conflict of interest to be addressed. If a conflict of interest exists, this should be mentioned in accordance with the journal guidelines and guidelines of good research practice.
Comments on the Quality of English LanguageEnglish generally fine, minor errors
Author Response
REPLY TO REVIEWS
Dear Editor and Reviewers,
Thank you very much for your help and input. Our answers are in italics.
Best regards
The authors
REVIEWER 2
The article “Creation of the corneal flap for laser in situ keratomileusis with a 3-dimension femtosecond laser cut: clinical and optical coherence tomography features” is an early study to validate and compare the new 3D refractive software of the Ziemer Z8 laser in a retrospective cohort. The study’s aim/novelty is to collect preliminary evidence on the new program for use of flap design in order to facilitate higher powered, prospective studies in the future. This can generally be considered a legitimate reason to conduct a clinical study. Please make the research hypothesis and questions more clear in the present manuscript.
The research hypothesis has been added and clarified (page 3-4, lines 49-51 and 59-66): assessment of reliability of the achieved angle, complications.
It is evident from the language used that the authors have committed to a surgical technique and technical system. Whether it is adequate to promote their technique (call it “gold standard”) and feature the technical system is to some degree left in the responsibility of the authors.
We mitigated our statement on “gold standard” and added a reference to prove that LASIK is the most used correcting technique (page 2, lines 19-20).
The reporting of the study could be improved in terms of organisation and logic, please consider relevant reporting guidelines.
Organisation and logic were improved all over the paper (in particular, page 3, lines 38-51 and 59-61; page 4, lines 63-66; page 9, lines 174-184).
The images appear a bit isolated in the manuscript; it would be better to group and arrange relevant images in a logic way as a panel. They could also be made a bit smaller.
The images have been grouped where appropriate, and made smaller.
It is recommended that the refractive software which seems important for the present study is better described. Specifically, the following questions should be answered: When was the 3D software released? What are its capabilities (including novelties compared to other suites on the market)? How does the updated version differentiate from previous versions (such as the 2D model, optima)? It would be good to illustrate the software with a telling screenshot, image (or even video) which demonstrates its functionalities. In addition, is the capability of the software really unique or can manual techniques accomplish a similar result? It is important to link the functionality of the software to the results achieved in the present clinical study (how does the software produce these results?). This is very central if the aim is to conduct future studies on the topic. The output of the present research project should be the generation of new hypothesis which subsequently can be addressed in follow-up studies in an improved research setting. Please specify these newly generated hypothesis to generate a basis for future research.
All the questions arisen have been now answered (page 6, lines 107-114). A new figure (4) has been added as suggested to introduce the software functions.
On a technical/methodological level, would it be interesting to follow the cases with aberrometry or are the results unlikely to be different from other surgical approaches?
This is a good input for future research, which we will investigate further. This has been added (page 11, lines 219-220).
Refractive surgeons often have relationships to the industry due to the technical nature of the discipline which requires the question of a conflict of interest to be addressed. If a conflict of interest exists, this should be mentioned in accordance with the journal guidelines and guidelines of good research practice.
We are independent researchers, with no conflict of interests, as stated at page 21.
Reviewer 3 Report
Comments and Suggestions for Authors
The manuscript ID optics-2945977 mainly presents a study about a particular optical coherence tomography 3D and 2D flaps. The article considers intra- and early post-operative complications in the study. Please see below a list of comments to the authors:
1. Please comment how the presented results can be confronted with updated publications related to the topic.
2. The advantages and disadvantages LASIK flap by a 3D femtosecond laser should be highlighted and better described in the discussion section.
3. It would be useful for readers if the authors describe how are formed the parallel fringes illustrated in figures 6 and 7.
4. The reviewer is curious if there is a potential influence of the incident polarization of the femtosecond pulses in the experiment.
5. Is it representative the change in absorption at the wavelength employed considering a different eye color? Please comment.
6. Perspectives can be described with better details. The authors are invited to describe some aspects of modern tools that can be useful for future research like the assistance of machine learning or controlled multiphotonic phenomena in surgical correction as you can see for instance: https://doi.org/10.3390/bios12090710
7. A graphical abstract would be welcome to better visualize the justification of the study, originality, and what this work adds to literature.
8. Besides the list of references could be updated, in my opinion, the introduction should be importantly improved to describe the panoramic aspects that motivate the aim of the work and the progress of the topic.
9. It is suggested to separate the discussions and conclusions in different sections; this is in order to easily see the value of the main findings.
10. Could you provide the characteristics of the femtosecond laser employed like pulse duration and pulse repetition rate in the manuscript for the audience of this prestigious journal Optics? This is in order to compare the effect in respect to other laser systems, you can see a different example, doi: 10.5455/medarh.2021.75.204-208
Author Response
REPLY TO REVIEWS
Dear Editor and Reviewers,
Thank you very much for your help and input. Our answers are in italics.
Best regards
The authors
REVIEWER 3
The manuscript ID optics-2945977 mainly presents a study about a particular optical coherence tomography 3D and 2D flaps. The article considers intra- and early post-operative complications in the study. Please see below a list of comments to the authors:
- Please comment how the presented results can be confronted with updated publications related to the topic.
Now added (page 9, lines 177-184).
- The advantages and disadvantages LASIK flap by a 3D femtosecond laser should be highlighted and better described in the discussion section.
Added (page 3, lines 44-51, and page 9, lines 167-173).
- It would be useful for readers if the authors describe how are formed the parallel fringes illustrated in figures 6 and 7.
Done (figure legends).
- The reviewer is curious if there is a potential influence of the incident polarization of the femtosecond pulses in the experiment.
We could not assess this aspect, due to the prevalently clinical aspect of the study.
- Is it representative the change in absorption at the wavelength employed considering a different eye color? Please comment.
The infrared femtosecond laser is focused within the corneal plane and its absorption by the iris is irrelevant.
- Perspectives can be described with better details. The authors are invited to describe some aspects of modern tools that can be useful for future research like the assistance of machine learning or controlled multiphotonic phenomena in surgical correction as you can see for instance: https://doi.org/10.3390/bios12090710
Perspectives have been described in more details (page 11, lines 217-233). We apologise, but we have no expertise in machine learning or multiphotonic phenomena.
- A graphical abstract would be welcome to better visualize the justification of the study, originality, and what this work adds to literature.
Due to our subspecialty features, we preferred to enhance the justifications of the study, originality, etc. in the fully rewritten Introduction.
- Besides the list of references could be updated, in my opinion, the introduction should be importantly improved to describe the panoramic aspects that motivate the aim of the work and the progress of the topic.
References have been updated, with 10 new quotations. The aim and progress are the subjects of the new Introduction.
- It is suggested to separate the discussions and conclusions in different sections; this is in order to easily see the value of the main findings.
Done (page 11).
- Could you provide the characteristics of the femtosecond laser employed like pulse duration and pulse repetition rate in the manuscript for the audience of this prestigious journal Optics? This is in order to compare the effect in respect to other laser systems, you can see a different example, doi: 10.5455/medarh.2021.75.204-208
The characteristics were added (page 6, lines 112-114).
Round 2
Reviewer 2 Report
Comments and Suggestions for Authors
Thank you for implementing the corrections; the manuscript is now more complete. As a side note, the manuscript's numbering of the pages and lines is different in the author's and reviewer's version thus I could not directly follow the changes based on the line numbers. For the addition of the hypothesis, I assumed that this was done in line 79-82 in my version (end of Introduction section).
I could not confirm any changes in the arrangement of the images, but I am assuming that these will be executed in the final version. Concerning the newly added images (specifically Figure 4), the centration of the screen photographs is not optimal (bad alignment); a higher quality image would be preferred, and for consistency it may be better to add arrows to the images as well and explain in the caption.
In-text correction:
L40: "thus rendering the induced benefits unpredictable" instead of "thus rendering unpredictable the induced benefits"
Comments on the Quality of English Language
Minor English check required on newly implemented corrections
Author Response
We confirm that the hypothesis is at the end of the introduction.
We improved figure 4 alignment and quality, and added arrows to explain various features, as specified in the caption.
L40 has been corrected as suggested.
Thank you very much for your help.
Reviewer 3 Report
Comments and Suggestions for Authors
I agree with the reviewed version of the report. In my opinion, this work can be considered for publication in present form.
Author Response
We thank you very much for your help and approval.